# Fraser Syndrome: A Narrative Review Based on a Case from Vietnam and the Past 20 Years of Research

**DOI:** 10.3390/diagnostics15131606

**Published:** 2025-06-25

**Authors:** Xuan Trang Thi Pham, Phuc Nhon Nguyen, Xuan Song Hoang

**Affiliations:** 1Department of Pathology Pregnancy, Tu Du Hospital, 284 Cong Quynh Street, Pham Ngu Lao Ward, District 1, Ho Chi Minh City 700000, Vietnam; xuantrangdr@yahoo.com; 2Clinical Research Center, Tu Du Hospital, Ho Chi Minh City 71012, Vietnam; 3Service de Gynécologie-Obstétrique, Maternité, Chirurgie Gynécologique, Centre Hospitalier Universitaire d’Orléans, 45100 Orléans, France; 4Department of General Surgery, People’s Hospital 115, Ho Chi Minh City 700000, Vietnam

**Keywords:** cryptophthalmos, gene mutation, syndactyly, Fraser syndrome, ultrasound

## Abstract

**Introduction**: Fraser syndrome (FS) is a rare autosomal recessive disorder. However, the clinical presentation remains variable. Diagnosis is based on a series of major and minor clinical criteria that can be supported by genetic tests. Prenatal diagnosis remains challenging. **Methods**: Herein, we reported a case of Fraser syndrome that was missed by ultrasound and diagnosed late at birth. The newborn presented with cryptophthalmos–syndactyly syndrome and absence of the right kidney. Based on a literature review of articles from the past 20 years, the authors found 40 cases, including indexed cases on PUBMED, Scopus, Web of Science, and Scholar using keywords related to “Fraser syndrome”. Through this report, we discuss the polymalformative syndrome, the clinical and paraclinical aspects of this syndrome, its clinical management, and highlight the importance of prenatal diagnosis in the light of research. **Results**: Our study found that consanguine parents (41.0%) were increasing risk factors for FS and poor socio-economic status delayed the early detection of FS. Among the 40 cases, 27 cases were detected postnatally. More than half of the cases resulted in poor perinatal outcomes. The common findings were cryptophthalmos (87.5%), syndactyly (87.5%), renal abnormalities (55.5%), and genital abnormalities (42.5%). **Conclusions**: A prenatal diagnosis of Fraser syndrome is still difficult. Thus, a counseled ultrasound scan at a specialized center should be recommended in suspected cases with indirect signs and risk factors of consanguinity.

## 1. Introduction

Fraser syndrome (FS; cryptophthalmos–syndactyly syndrome; OMIM #219000), also known as Meyer-Schwickerath syndrome, Fraser–François syndrome, or Ullrich–Feichtiger syndrome, is a rare autosomal recessive multiple malformation syndrome characterized by cryptophthalmos, syndactyly, and respiratory and urogenital tract anomalies [1,2]. The prevalence of FS is 0.43 per 100,000 among newborns worldwide. Since the first description of the syndrome by George Fraser in 1962 and later by Thomas et al. in 1986, over 280 cases have been documented to date [3,4].

Fraser syndrome is caused by mutations in the FRAS1 (4q21.21), FREM2 (Fras1-related extracellular matrix gene 2) (13q13.3), and GRIP1 (glutamate-receptor-interacting protein 1) (12q14.3) genes, coding for extracellular matrix proteins essential for the adhesion between the basement membrane of the epidermis and connective tissues of the dermic layer during embryological development. Mutations in these genes are suggested to be responsible for failure in apoptosis [5,6].

Risk factors include consanguine marriage and familial factors [7]. Consanguineous marriage can increase the risk of autosomal recessive conditions in general. Van Healst et al. reported 59 affected individuals from 40 families. Among them, 25 cases were consanguineous [8]. According to Slavotinek et al., FS was present in more girls (57/117) than boys and consanguinity was present in 29/119 (24.8%) of the cases [8]. Cryptophthalmos and syndactyly seem to be the most constant features of this disorder, with cryptophthalmos being the most defining feature but not an obligatory feature of Fraser syndrome [2,9,10]. Other anomalies include CHAOS (congenital obstructive syndrome of the upper airways). A wide variability of expression has been observed, making its clinical diagnosis sometimes challenging [11]. Many cases were detected postnatally [12,13,14,15,16,17].

Through this report, the team emphasizes the characteristics of Fraser syndrome. In addition, we searched English keywords relating to Fraser syndrome, including “Fraser syndrome”, or “Meyer-Schwickerath’s syndrome” or “Fraser-François syndrome”, or “Ullrich-Feichtiger syndrome” and “case report” on databases such as Medline, PubMed, Scopus, and Google Scholar to review the literature from the past 20 years.

## 2. Case Presentation

A 29-year-old Vietnamese pregnant woman (gravida: 0, parity: 0) with a non-consanguineous Vietnamese partner was hospitalized for induction of labor at 38 weeks and 4 days due to fetal growth restriction (FGR). There was no reported history of toxoplasma, cytomegalovirus (CMV), or rubella infections. Additionally, there was no history of radiation exposure, drug use, and no reported maternal history of fever or genitourinary infections during the pregnancy. There was also no history of cryptophthalmos or other significant abnormalities in their immediate or distant families. The maternal blood group was A with a positive rhesus factor. Both the mother and father were non-smokers. The mother did not consume alcohol.

During the first trimester screening, the nuchal translucency measurement was normal. A non-invasive prenatal test (NIPT) showed a normal karyotype. Trisomy 13, 18, 21, and sexual chromosomal aneuploidy (monoX, XXY, XXX, XYY, XXXY) were low-risk. The screening test for nine common recessive genetic diseases in maternal plasma, including genetic mutations of HBA1 and HBA2 (Alpha thalassemia), HBB (Beta thalassemia), GALT (metabolic disorder of galactose), PAH (urine phenylketone), G6PD (glucose-6-phosphate dehydrogenase deficiency), SLC25A13 (solute carrier family 25 member 13), SRD5A2 (lack of 5-alpha reductase), GAA (metabolic disease of glycogen type 2, Pompe disease), and ATP7B (metabolic disorder of Cu, Wilson disease), were absent.

A four-dimensional morphologic ultrasound at 21 weeks and 5 days was performed without abnormal detection, except for a fetal growth restriction (FGR) with an estimated fetal weight (EFW) below the third percentile. The head circumference measured at 174 mm, abdominal circumference measured at 151 mm, and femoral length measured at 32 mm. The fetal heart rate and fetal movement were normal. The amniotic fluid volume was measured at 6–7 cm. The patient was counseled for amniocentesis; however, she refused.

At 36 weeks of gestational age (GA), the ultrasound showed a small gestational-age fetus with an EFW of 2270 g (5.7% following Hadlock, corresponding to 33 weeks and 4 days of gestational age). The biometric parameters included a biparietal diameter (BPD) of 81 mm (3%), head circumference (HC) of 306 (1%), abdominal circumference (AC) of 297 mm (5%), and femoral length (FL) of 65 mm (2%). The amniotic fluid index (AFI) measured at 10 cm. During pregnancy, the patient was followed up at a different hospital.

During hospitalization, the patient’s cardiotocography was normal. Since the induction of labor failed after using both a balloon and prostaglandin E2, the patient underwent cesarean delivery. The cesarean section was performed uneventfully with a female newborn weighing at 2700 g. The Apgar score was evaluated at 6/1 min and 8/5 min.

However, the neonate was observed to have abnormal morphology. Unilateral cryptophthalmos and cleft palate were noted postnatally. In addition, the neonate presented with abnormalities of the extremities, consisting of partial III-IV-V finger membranous syndactyly on the right hand, partial II-III-IV finger membranous syndactyly on the left hand, and complete II-III-IV-V foot membranous syndactyly on the right side, and complete IV-V membranous syndactyly on the left side. The face was dysmorphic, with marked hypertelorism, a depressed nasal bridge, dysplastic ears, and low-set ears (Figure 1A–G). The rest of the systemic examination was unremarkable.

Based on the clinical aspect, the neonate was suspected of Fraser syndrome according clinical criteria of Thomas and Van Healst (Table 1).

After birth, the baby was strictly monitored at the neonatal unit for 5 days. Both mother and neonate were discharged on day 5 after birth. The postnatal ultrasound scan demonstrated right renal agenesis. Meanwhile, anterior to the right iliopsoas muscle, a cystic structure measuring 9.5 × 7.3 mm was suspected to be right renal ectopic dysplasia. The infant gained 3300 g by the 28th day of life. The neonate was monitored for one month with the same malformation at birth and without other complications of the digestive tract and cardio-pulmonary system (Figure 2A–G) and Appendix A.

The patient displayed total cryptophthalmos of the right eye and an absent eyebrow but the ocular globe was felt underneath the skin. The diagnosis was confirmed by genetic analysis, which showed novel compound heterozygous variants of FRAS1 gene on chromosome 4 was found. Concisely, 03 heterozygous variants not reported in ClinVar (classified as likely pathogenic (c.4426-1G>A, location 78418948) and uncertain significance (c.4426G>C, location 78418949 và c.11411A>G, location 78539406) according to the American College of Medical Genetics and Genomics (ACMG) criteria) in the FRAS1 gene (Figure 3). Thus, pathogenic/likely pathogenic homozygous or in-trans compound heterozygous variants in this gene may be associated with Fraser syndrome 1; probably linking to clinical diagnosis. The parental genetic analysis revealed a heterozygous variant of the FRAS1 gene on chromosome 4, location 78418948, relating to a variation of NM_025074.7:c.4426-1G>A (Father) and a heterozygous variant of the FRAS1 gene on chromosome 4, location 78539406, relating to a variation of NM_025074.7:c.11411A>G (Mother) (Appendix A).

Until today, the baby was monitored for a systematic approach of surgical management in the future.

## 3. Discussion

Fraser syndrome (MIM#219000) is an autosomal recessive disorder characterized by the association of cryptophthalmos, syndactyly of the four extremities, urinary tract abnormalities, and laryngo-tracheal anomalies. This condition is due to homozygous or compound heterozygous mutations in the FRAS/FREM complex genes: FRAS1, FREM2, and GRIP1. Severe malformations have not been reported in association with the type of gene mutation. However, GRIP1 gene mutation was documented in severe forms and the fetus died early [6]. Consanguinity is a risk factor for autosomal recessive disorders. Parents with a low socioeconomic background and limited access to medical facilities are contributing factors for delayed detection of FS. Due to genetic counseling or fear of repeated FS, almost all cases were primipara. Previously, one case was recorded during the ninth pregnancy. Additionally, in one case, FS was reported during the second pregnancy and the fourth pregnancy; the third pregnancy was normal (Table 2).

The diagnosis of FS is based on clinical findings, imaging, and genetic tests for causative mutations [1]. Karyotype and imaging of the urogenital tract may be required in case of ambiguous genitalia. The diagnostic criteria for FS are divided into six major criteria (syndactyly, cryptophthalmos spectrum, urinary tract abnormalities, ambiguous genitalia, laryngeal and tracheal anomalies, and positive family history) and five minor criteria (anorectal defects, dysplastic ears, skull ossification defects, umbilical abnormalities, nasal anomalies) [8]. However, major manifestations such as cryptophthalmos and syndactyly are difficult to detect through prenatal ultrasonography, especially in the presence of oligohydramnios. Similarly, a prenatal ultrasound did not reveal or raise any suspicions for Fraser Syndrome in the present case. Only a few prenatal cases have been reported. The earliest diagnosis in our findings was between 18–19 weeks of GA. Almost all cases were diagnosed late after birth. The latest case was diagnosed 9 months postnatally. The diagnosis is usually made at birth from the obvious malformations, although it can also be diagnosed during prenatal ultrasounds. Therefore, the patient should be counseled at a consultant center when detecting one of the anomalies on ultrasound, such as ophthalmic abnormality or renal agenesis. Additionally, some indirect signs could help to investigate the malformation. For example, severe oligohydramnios may be caused by renal agenesis or hyperechoic lung and ascites in congenital obstructive syndrome of the upper airways (CHAOS) (Table 3 and Table 4). In addition, magnetic resonance imaging (MRI) could be indicated where necessary.

Generally, Fraser syndrome is highly variable, its malformations ranging from minor signs to lethal anomalies. Mostly, FS is usually characterized by eye (unilateral or bilateral cryptophthalmos, microphthalmos/anophthalmos), urinary tract (renal agenesis, bladder atresia/hypoplasia), and genital anomalies (ambiguous genitalia, cryptorchidism, underdevelopment of male genitalia). Cutaneous syndactyly occurs in both hands and feet. Patients may also present craniofacial (dysplastic ears, bifid nose, cleft lip/palate, microglossia), respiratory (laryngeal stenosis/hypoplasia), cardiac (atrial and ventricular septal defect), digestive tract (anal agenesis/imperforation, diaphragmatic hernia, large bowel obstruction), and skeletal anomalies (skull/spine malformations, clubfoot) [18,19]. In the case of severe manifestations, the disease may be fatal for the fetus; the major causes of early death after birth include laryngeal and/or kidney anomalies. According to Slavotinek et al., cryptophthalmos was present in 103/117 (88%), syndactyly in 72/117 (61.5%), and ambiguous genitalia in 20/117 (17.1%) of the cases. Ear malformations were recorded in 69/117 (59%) and renal agenesis in 53/117 (45.3%) of the cases [1]. The results were similar to our findings in the literature review of the past 20 years (Appendix A). Other abnormalities include umbilical hernias, a single umbilical artery, hydrometrocolpos secondary to vaginal atresia, and imperforate hymens [20].

Clinically, cryptophthalmos is defined as the continuous passage of skin from the forehead to the cheek and the absence of a palpebral fissure. It may be related to microphthalmia and anophthalmia [21]. Cryptophthalmos may be unilateral or bilateral and may be found in isolation or as one component of a syndrome. Upon high-resolution ultrasound examination, skin was seen to be continuous over the eyeballs on both sides and the palpebral fissure could not be identified. Echoes from the lens and ciliary apparatus could not be seen within the eyeballs, which revealed a single cavity [22]. Possible differential diagnoses should be made, which include ankyloblepharon (partial or complete adhesion of the ciliary edges of superior and inferior eyelids), blepharophimosis (narrowing of the eye opening), and symblepharon (partial or complete adhesion of the palpebral conjunctiva of the eyelid to the bulbar conjunctiva of the eyeball) [23]. Syndactyly is defined as the fusion of two or more digits (bone or soft tissue). It can be present in other syndromes, such as Apert syndrome, Carpenter syndrome, Manitoba oculotrichonal syndrome, and Lenz microphthalmia syndrome [24,25]. Relating to this abnormality, the patient is usually diagnosed after birth at an ophthalmology department or plastic surgery outpatient clinic [2,26,27,28].

Accordingly, treatment is dependent on which malformations are present, and genetic counseling is indicated. Emergency surgery is required in the case of respiratory distress [29]. Following functional stability, esthetics are considered. Management requires many step-by-step surgeries [2]. The surgical repair of FS requires a multidisciplinary team (including a maxillofacial surgeon, ear–nose–throat (ENT) specialist, nephrologist, ophthalmologist, and other specialists) [30,31]. The prognosis is dependent on the severity of the renal and laryngeal malformations. Noticeably, CHAOS is a severe abnormality [32].

To date, survival rates of FS depend on the severity of the associated anomalies, with laryngeal and/or kidney malformations as the major causes of fatality [33]. Development could be delayed or normal with the survival rate varying from 2 to 32 years. Previously, there were some reports of people surviving beyond the age of 20 years [3,31]. Some other defects could be present in the adolescent along with other abnormalities, such as craniosynostosis, hearing loss, and generalized tonic–clonic seizures [34].

**Table 2 diagnostics-15-01606-t002:** Pregnancy characteristics of Fraser syndrome in the literature of the past 20 years.

Reports (Authors, Published Year, Country)		GA at Detection	Age of Parent	Pregnancy Characteristics	History	Risk Factors	Newborn’s Gene Mutation	Parent Gene Mutation	Outcomes
Vijayaraghavan et al. (2005), India [22]		Antenatal at 26 weeks	24 yo	G2P1	First baby died on 2nd day due to congenital heart disease	-	-	-	Termination of pregnancy
Allali et al. (2006), Casablanca [35]		Postnatal	24 yo	G2P2	First newborn died after birth	Consanguineous marriage	-	-	- SVD - Term - Alive
Slavotinek et al. (2006), USA [33]	Case 1	Postnatal	-	-	-	-	c.5446delTCTTTC in exon 40 and c.6992insGG in exon 49	-	Survival of 2 years
	Case 2	Postnatal	14 yo	-	-	Unknown history of consanguinity or exposures during the pregnancy.	c.G3095A, predicting p.G1032E in heterozygous form in exon 24	-	- SVD at 42 weeks of GA - Survival at 13 years
Kumari et al. (2008), India [32]		Postnatal	-	First child	-	Second-degree consanguineous parents	-	-	- CS at full-term - Death
Eskander et al. (2008), USA [12]		Postnatal	28 yo	Primipara	-	-	-	-	IOL for postdate CS due to failed IOL Died after birth due to ARI.
Dilli et al. (2010), Turkey [18]		Postnatal	27 yo	G5P4	-	-	-	-	Male 2420 g 46 cm Poor outcome
Yassin et al. (2010), Sudan [7]		Postnatal	25 yo	Second child	Healthy	Consanguineous parents paternal cousins had FS No regular antenatal US	-	-	SVD at term
Pasu et al. (2011), England [14]		Postnatal day 1	-	Primipara	-	Non-consanguineous parents of Southeast Asian origin.	-	-	- Vaginal ventouse delivery at 39 + 2 weeks - Male, 2.8 kg - Alive
Vogel et al. (2012), Netherlands [6]	Case 1	Prenatal	-	Third pregnancy	-	-	- Mutation analysis of GRIP1 revealed that Chr12 (GRCh37):g.66786456C/G, NM_021150.3:c.2113 + 1G/C substitution - Homozygous for the mutation	the parents and two unaffected siblings were heterozygous.	Stillborn 20 weeks
	Case 2	Prenatal	-	-	-	Consanguineous	-	-	Stillborn boy after 30 weeks
	Case 3	Prenatal	-	-	-	Consanguineous	-	Heterozygous for a 4-bp deletion in exon 10 of GRIP1 (NM_021150.3:c.1181_1184del) (p.Lys394ThrfsX8)	Pregnancy termination at 30 weeks
Hoefele et al. (2013), Germany [36]		Prenatal at 11 weeks	-	G2P0	Stillborn at 29 weeks, suspicion of FS	Healthy	Mutation of FRAS1 gene	Both parents are heterozygous for a FRAS1 mutation.	Abortion at 13 weeks
Lorena et al. (2014), Brazil [26]		Postnatal at 9 months	-	-	-	Consanguineous	-	-	Female child Alive
De Bernardo et al. (2015), Italy [29]		Postnatal	-	-	-	-	Mutations c.[5752dup]; [8544 + 1G>T] p.[(Cy51918fs)], in the gene FREM2	-	- Natural birth at 40 weeks, 3 d - Female - 3220 g - 46 cm - Apgar score 4/6 - Tracheostomy
Saleem et al. (2015), Pakistan [37]		Postnatal at 3 months	-	Sixth pregnancy	Healthy	Consanguineous	Genetic analysis was not done due to financial constraints	-	-SVD at home - 2 kg
Sangtani et al. (2015), India [2]		Postnatal	-	Second child	Healthy	Consanguineous	-	-	SVD Term Male
Abdalla et al. (2016), Egypt [11]	Case 1	Postnatal at 6 months		Second child	Healthy	Consanguineous	-	-	Female
	Case 2	Postnatal	-	First child	Healthy	Consanguineous	-	-	Term Death after birth
	Case 3	Antenatal at 28 weeks	30 yo	Second child	Healthy	Consanguineous	-	Normal karyotype	Termination Male
	Case 4	Postnatal	-	Pregnancy after 7 first trimester abortions	-	Non-consanguineous	-	-	Full term Stillborn Male
	Case 5	Antenatal at 24 weeks	-	Pregnancy after 7 first trimester abortions and full-term with FS	-	Non-consanguineous	-	-	Termination
	Case 6	Antenatal at 24 weeks	21 yo	Primigravida	Healthy	-	-	-	Termination
Selvaraj et al. (2016), India [38]		Antenatal at 22 weeks	22 yo	G4P4	- First pregnancy was terminated at 24 weeks due to suspicion of bladder outlet obstruction and ascites. - Second pregnancy was diagnosed with FS	- Consanguineous marriage - History of pregnancy with FS	-	-	Termination of pregnancy
Dumitru et al. (2016), Romania [13]		Postnatal	-	G2P2	Healthy parents	Non-consanguineous marriage	-	-	- CS - Female -1480 g - 38 cm - Death due to ARI
Dar Parvez Mohi Ud Din et al. (2017), India [39]		Postnatal at 6 months	-	Second child	-	Non-consanguineous marriage	Mutation of the FRAM2	-	- CS at 40 weeks - 2.8 kg
Mbonda et al. (2019), Cameroon [24]		Postnatal at 6 months	29 yo	Primipara	- HIV - Psychiatric disorders - Bilateral club foot.	- No antenatal care - No consanguinity - Separated parents	-	-	- SVD at term of 9 months - Female - Severe illness at age of 6 months due to pneumonia.
Alsaman et al. (2020), Syria [25]		Postnal	19 yo	G2P1	-	Non-consanguineous marriage	-	-	- CS at 39 weeks due to cephalopelvic disproportion - 3600 g - Death due to respiratory insufficiency
Ikeda et al. (2020), Japan [4]		Antenatal	38 yo	G2P0	-	Non-consanguineous marriage	-	-	- Terminated pregnancy at 21 weeks - Male - 520 g - 29.5 cm
Waseem et al. (2021), Pakistan [28]		Posnatal at 2 months	-	Primipara	-	- Consanguineous - Low socioeconomic background	-	-	SVD Term Male 3 kg
Shrestha et al. (2022), Nepal [40]		Posnatal	24 yo	-	-	- Low socioeconomic condition - Non-consanguineous	-	-	- SVD - 1400 g - 45 cm - Apgar score of 3 pts at 1 min and 4 pts at 5 min. - Death after 3 h of birth.
Golshahi et al. (2022), Iran [41]		Antenatal at 18 weeks and 3 days	26 yo	Primipara	-	- Consanguineous marriage - No pregnancy follow-up	Absent due to high expenses	- Heterozygous mutations in the FRAS1 gene - Some other genes, including NBAS, MYL3, PKD1L1, and NUP160	- Abortion with PGE1 - Death 600 g
Laminou et al. (2022), Nigeria [15]		Postnatal day 1	-	Sixth pregnancy	-	- Consanguineous marriage - Poorly monitored	Refused	-	- SVD - Term - Female - 2800 g - 49 cm
Neri et al. (2023), Colombia [20]		Antenatal at 35 weeks	19 yo	Primipara	Healthy	Non-consanguineous marriage	-	-	- CS for breech presentation - 36 weeks - Female - 2550 g - Apgar 2/1 min and 0/5 min - Death
Ramadugu et al. (2023), India [23]		Postnatal	23 yo	Primipara	High blood glucose levels during 6 to 7 months of pregnancy	No history of consanguineous marriage	Lack of resource settings	-	- CS - female - 3416 g - NICU admission for 2 days due to ARI. - Alive
Bouba Traore et al. (2024), Nigeria [19]	Case 1	Postnatal day 15	22 yo	Second child	Healthy	-	-	-	SVD Male Term 3200 g 50 cm Alive
	Case 2	Postnatal day 3	34 yo	Eighth child	Healthy	Consanguinity	-	-	SVD Female Term 2980 g 51 cm Alive
Mohamed et al. (2024), Sudan [42]		Postnatal	22 yo 24 yo	-	Malaria and urinary tract infection during pregnancy	- Poorly Monitored - Non-consanguineous marriage	-	-	Female
Sajoura et al. (2024), Morocco [16]		Postnatal day 1	34 yo 35 yo	Primiparous	Healthy	- Non-consanguineous marriage - Poor follow-up during pregnancy	-	-	CS at 42 weeks Death at the age of 40 days due to severe ARI + hypovolaemic shock
Mangla et al. (2024), India [10]		Antenatal	21 yo	Primigravida	-	-	-	-	Pregnancy termination
Present case (2015), Vietnam		Postnatal	29 yo	Primipara	Healthy	No	Heterozygous variation of FRAS1 gene on chr4, location 78539406, variants NM_025074.7:c.11411A>G (NP_079350.5:p.Asp3804Gly)	Parental gene analysis revealed a heterozygous variant of FRAS1 gene on chromosome 4, location 78418948 and 78539406	- IOL for SGA - CS due to failed IOL Alive

ARI: acute respiratory insufficiency; CS: cesarean section; d: days; G: gravida; g: gram; GRIP1: glutamate receptor interacting protein 1; IOL: induction of labor; kg: kilogram; P: parity; SGA: small gestational age; SVD: spontaneous vaginal delivery; FRAS1: Fraser extracellular matrix complex subunit 1; FREM2: FRAS1 related extracellular matrix protein 2. Sign “-”: not mentioned; weeks: weeks; yo: years old.

**Table 3 diagnostics-15-01606-t003:** Cases of prenatal diagnosis of Fraser syndrome using ultrasound in literature of the past 20 years.

Reports (Authors, Published Year, Country)		Time Point	Cryptoph Thalmos	Syndactyly	Urinary Abnormality	Respiratory Abnormality	Ear Malformation	Genital Abnormality	Others
Vijayaraghavan et al. (2005), India [22]		26 weeks	+ Unilateral microphthalmos Skin over the eyeballs on both sides and unidentified palpebral fissure	+ Bilateral cutaneous syndactyly of all fingers	+ Agenesis of the right kidney	-	+ Small ears and deformed	+ Short penis and lacked preputial cover	SGA small omphalocele
Allali et al. (2006), Casablanca [35]		-	-	-	-	-	-	-	-
Slavotinek et al. (2006), USA [1]	Case 1	-	-	-	-	-	-	-	-
	Case 2	-	-	-	-	-	-	-	-
Eskander et al. (2008), USA [12]		-	-	-	-	-	-	-	-
Kumari et al. (2008), India [32]		-	-	-	-	-	-	-	Severe intrauterine growth retardation Oligohydramnios
Dilli et al. (2010), Turkey [18]		-	-	-	-	-	-	-	-
Yassin et al. (2010), Sudan [7]		-	-	-	-	-	-	-	-
Pasu et al. (2011), England [14]		20 weeks	-	-	+ Unilateral right renal agenesis	-	-	-	-
Vogel et al. (2012), Netherlands [6]	Case 1	-	-	-	-	+ Tracheal atresia	-	-	Severely hydropic fetus and ascites.
	Case 2	-	+ Cryptophthalmos		+ Absence of bladder and kidneys	+ Hyperechogenic lungs	-	-	-
	Case 3	-	-	-	+ Bilateral renal agenesis	-	-	-	Severe oligohydramnios
Hoefele et al. (2013), Germany [36]		11 weeks	-	-	-	-	-	-	Bilateral cleft lip and palate.
Lorena et al. (2014), Brazil [26]		-	-	-	-	-	-	-	-
De Bernardo et al. (2015), Japan [29]		-	-	-	-	-	-	-	-
Saleem et al. (2015), Pakistan [37]		-	-	-	-	-	-	-	-
Sangtani et al. (2015), India [2]		-	-	-	-	-	-	-	-
Selvaraj et al. (2016) [38]	Case 1	22 weeks	-	-	+ Unilateral renal agenesis	+ Bilateral hyperechoic enlarged lungs. Diaphragmatic inversion and flattening were seen. Congenital high airway obstruction	-	-	- Fetal heart was seen in midline, appeared compressed - Ascites - Single umbilical artery - Reduced amniotic fluid
	Case 2	13–16 weeks	-	-	+ Unilateral renal agenesis	+ Bilateral enlarged hyperechoic lungs CHAOS	-	-	- Single umbilical artery - Ascites - Transverse section of kidney
Dumitru et al. (2016), Romania [13]		-	-	-	-	-	-	-	-
Dar Parvez Mohi Ud Din et al. (2017), India [22]		-	-	-	-	-	-	-	-
Mbonda et al. (2019), Cameroon [24]		-	-	-	-	-	-	-	-
Alsaman et al. (2020), Syria [25]		20 and 25 weeks	-	+ Bilateral syndactyly on both hands and feet	-	+ Hyperechoic lungs Enlargement of the lungs	-	-	- Ascites - Shortness in upper and lower limbs, nuchal edema, hydrops fetalis - Cardiac compression - Intrahepatic biliary atresia
Ikeda et al. (2020), Japan [4]		19 weeks	-	-	+ Bilateral Renal agenesis, no urinary bladder,	+ Distended trachea from the caudal of the carina to the bronchi Enlarged lungs - CHAOS	-	-	Severe oligohydramnios, high volume of ascites Hepatomegaly
Waseem et al. (2021), Pakistan [28]		-	-	-	-	-	-	-	-
Shrestha et al. (2022), Nepal [40]		-	-	-	-	-	-	-	-
Golshahi et al. (2022) [41]		18 weeks 3 d	+ Unilateral microphthalmos	+ Bilateral syndactyly	+ Renal agenesis (found as a lying down the adrenal sign)	+ - Hyper-echogenic lungs - CHAOS	-	-	Single umbilical artery, severe oligohydramnios
Laminou et al. (2022), Nigeria [15]		-	-	-	-	-	-	-	-
Neri et al. (2023), Colombia [20]		36 weeks	+ Microphthalmia	-	-	-	-	-	Hypotelorism
Ramadugu et al. (2023), India [23]		-	-	-	-	-	-	-	Mild oligohydramnios
Bouba Traore et al. (2024), Nigeria [19]	Case 1	-	-	-	-	-	-	-	-
	Case 2	-	-	-	-	-	-	-	-
Mohamed et al. (2024), Sudan [42]		-	-	-	-	-	-	-	-
Sajoura et al. (2024), Morocco [16]		-	-	-	-	-	-	-	-
Mangla et al. (2024), India [10]		Antenatal	+ Absence of an eye globe and lens	-	+ Renal agenesis (sleeping adrenals sign) - Non-visualization of the urinary bladder, and Doppler of renal arteries.	-	-	-	- Severe oligohydramnios - Non-visualization of the urinary bladder
Present case (2025), Vietnam		-	-	-	-	-	-	-	- FGR - Reduced AFI

CHAOS: Congenital high airway obstruction syndrome; weeks: weeks; Sign “-”: not mentioned; Sign “+”: present.

**Table 4 diagnostics-15-01606-t004:** Postnatal characteristics among cases of Fraser syndrome in the literature of the past 20 years.

Report (Authors, Published Year, Country)		Eyes		Limbs	Urinary Tract	Respiratory Tract	Others
Vijayaraghavan et al. (2005), India [22]		Bilateral cryptophthalmos		Syndactyly of the fingers	Right kidney was absent	-	- Low frontal hairline - Bilateral low-set ears with abnormal pinna - Broad abnormal nose - Micrognathia - Micropenis - Small omphalocele containing a Meckel’s diverticulum.
Allali et al. (2006), Casablanca [35]		- Ankyloblepharon on the left side - Cryptophthalmos on the right side - A cornea reduced to an opaque thin plate clinging to the iris without an anterior chamber and a non-individualized eyeball		Syndactyly	Renal abnormalities	Laryngeal stenosis tracheal abnormality	- Anorectal abnormalities - Ambiguous genitalia - Ear malformations - CT scan of the cranium and orbits and the transfontanelle ultrasound were normal
Slavotinek et al. (2006), USA [33]		Case 1	Unilateral cryptophthalmos	No syndactyly	Hydronephrosis	Laryngeal stenosis/webs	- Tongue of hair extending from the anterior scalp hairline to the eyebrow - Asymmetry of the nares, stridor with minor webbing of the vocal cords, hearing impairment, hydronephrosis, and a bicornuate uterus - Inner ear dysplasia
		Case 2	Cryptophthalmos with small and fused palpebral fissures bilaterally, bilateral anophthalmia	-	-	-	- Bilateral cleft lip and palate. - An extension of the anterior hairline across the lateral forehead and the nares were hypoplastic with a groove to the left of the midline of the nasal tip and a right preauricular pit - Absence of the corpus callosum with abnormal folding of the gyri, partial fusion of the thalami, dysplasia of the hippocampi and lateral ventricles, and prominent caudate heads
Eskander et al. (2008), USA [12]		Bilateral cryptophthalmos		Upper and lower extremities demonstrated prominent soft-tissue syndactyly	Left kidney and ureter agenesis	Pulmonary hypoplasia	- A flat forehead - Small nose with depressed nasal bridge - External genitalia were ambiguous and show labial malformation with clitoral enlargement Vagina was atretic and anus was patent Large intestine showed markedly distended rectosigmoid area and left adrenal gland was slightly hemorrhagic
Kumari et al. (2008), India [32]		Complete cryptophthalmos of the right eye		Complete cutaneous syndactyly of both the hands and feet	Bilateral renal agenesis	Tracheal stenosis	The ears were malformed and low-set. The nose was flat with a wide nasal bridge. The testes were found at the pelvic brim.
Dilli et al. (2010), Turkey [18]		Bilateral cryptophthalmos		Syndactyly	Bilateral polycystic kidney	Laryngeal stenosis	- Flat nose, hypoplastic nose and ears, mycrostomy, bifid uvula and cleft palate - Perineal fistula and anal atresia - Ambiguous genitalia
Yassin et al. (2010), Sudan [7]		Bilateral cryptophtalmos		Syndactyly in both hands and feet	Bilateral dysplatic kidney		- Clitoromegaly with hypoplasia of the labia - Low-set umbilicus - Depressed nasal bridge, broad nose with midline nasal groove
Pasu et al. (2011), England [14]		Right complete cryptophthalmos was noted with a palpable eyeball beneath		Clinodactyly of the right fourth and fifth toes	-	-	- Umbilical hernia, widely spaced fontanelle, bulbous nose
Vogel et al. (2012), Netherlands [6]		Case 1	Bilateral cryptophthalmos	The fingers and toes were short with partial bilateral cutaneous syndactyly	There were no kidneys and a severely hypoplastic bladder.	The larynx was malformed and atretic; there was bilateral pulmonary hyperplasia with abnormal lung lobation.	- Low-set simple ears, micrognathia and a beaked nose with notched alae nasi - External genitalia were male with hypoplastic scrotum and malformed hypoplastic penis - Anus was abnormally positioned and appeared stenotic
		Case 2	Bilateral complete cryptophthalmos,	-	- Absence of autopsy	- Absence of autopsy	Abnormal frontal hairline and broad nose
		Case 3	-	-	Bilateral renal agenesis bilateral ureter agenesis hypoplasia of the bladder	-	Hydrocephaly and dysmorphic features typical for FS
Hoefele et al. (2013), Germany [36]		-		Syndactyly of the second to the fourth finger on the right hand and between the second and the fifth toe on both feet	Atresia of the epiglottis resulting in an occlusion of the trachea	-	-
Lorena et al. (2014), Brazil [26]		Total unilateral cryptophthalmos (left side), epiphora (right side) with mucopurulent discharge		Syndactyly of the fingers and toes	-	-	- Depressed nasal bridge, low-set ears, atresia of the external auditory canal, prominent labia majora - Brachycephaly, absent septum pellucidum, prominent lateral ventricles, major skull bone defect, thinning of the brain mantle, small posterior fossa
De Bernardo et al. (2015), Japan [29]		- Bilateral cryptophthalmos Occipitofrontal circumference 34.5 cm - Left microphthalmos and malformation-like coloboma into right ocular globe with cysts and small calcification parietal anterior		Syndactyly	Right kidney agenesis	- Sub-stenosis laryngeal - Respiratory distress	- Bradycardia - Bilateral microtia - Ambiguous genitalia - Incomplete myelination of the brain
Saleem et al. (2015), Pakistan [37]		- Bilateral cryptophthalmos - Microphthalmic		Bilaterally complete syndactyly	-	-	- Spaced nipples and an umbilical hernia - Genitalia were ambiguous with phallus, complete labial fusion, and absent testes - Hypertelorism
Sangtani et al. (2015), India [2]		-		Syndactyly of hands and feet,	Enlarged hydronephrotic left kidney, and dilated left ureter Right renal agenesis	-	Bilateral cleft lip and palate
Selvaraj et al. (2016) [38]		Case 1	-	- Partial syndactyly of fingers - Complete syndactyly of toes	-	-	- Abnormal genitalia with imperforate anus - Face was severely dysmorphic with marked hypertelorism, depressed nasal bridge, and low-set ears
		Case 2	-	- Partial syndactyly of 3rd and 4th fingers - Complete syndactyly of toes	-	-	- Abnormal genitalia with imperforate anus - Abortus showing wide open eyes with dysmorphic facies
Dumitru et al. (2016), Romania [13]		Complete bilateral cryptophthalmos		Complete bilateral syndactyly on both hands and feet	Bilateral renal agenesia with agenesis of the ureters, hypertrophic adrenal glands, hypoplasic bladder.	Hypoplastic lungs	- Severe facial dysmorphism with low-set ears, flat nasal bridge, micrognathia and incomplete ossification of the skull bones. - An umbilical hernia, imperforate anus, and ambiguous genitalia - Patent ductus arteriosus and foramen ovale.
Dar Parvez Mohi Ud Din et al. (2017), India [39]		- Normal anterior and posterior segment of the right eye with coloboma of the right upper eyelid - Left complete cryptophthalmos with a palpable eyeball beneath		Syndactyly of all fingers of bilateral hands and all toes of bilateral feet	Left renal agenesis	-	- Nasal deformity - Absence of bilateral testes in the scrotum
Mbonda et al. (2019), Cameroon [24]		Bilateral cryptophthalmos		Syndactyly	-	-	- Nasal malformation - Anal imperforation with a nearby anal fistula - External genitalia anomaly
Alsaman et al. (2020), Syria [25]		Bilateral anophthalmia		Bilateral syndactyly on hands and feet	Right renal agenesis	-Lung enlargement -Autopsy: Dilated pleural lymphatic vessels, interstitial fibroblast hypertrophy and vascular wall thickening in the lung.	- Pseudo-hypertelorism - Low-set ears, flat nasal bridge - Cutaneous and subcutaneous edema, large-volume ascites - Ambiguous genitalia - Gonad and Mullerian structures were found on the left posterior pelvic wall - Autopsy: Congestion of red pulp with hemosiderin accumulation and immature white pulp of the spleen
Ikeda et al. (2020), Japan [4]		Bilateral cryptophthalmos		Syndactyly of both hands (I–IV)	Autopsy: Agenesis of the kidneys, ureters, and bladder.	Autopsy: Pleural effusion and ascites, atresia of the epiglottis with a dilated trachea, hyperinflated and heavy lungs.	Low-set, malformed ears
Waseem et al. (2021), Pakistan [28]		- Complete cryptophthalmos in the right eye with partially formed eyebrow and absence of eyelids and eyelashes - Conjunctivitis in the left eye		Cutaneous syndactyly of hands	-	-	- Hypertelorism - Low-set ears - High-arched palate with ankyloglossia - Cryptorchidism and hypospadias
Shrestha et al. (2022), Nepal [40]		Absent eyelashes		Syndactyly of toes of both feet	-	-	- Wide anterior fontanelle - Ambiguous genitalia with phallus and complete labial fusion - Abnormal face bones, flat nasal bridge, prominent occiput, hypertelorism, and dysplastic low-set ears - Single umbilical artery - Sacral dimpling with tuft of hair and protrusion of tail-like appendages at back in sacrum; likely lipomeningocele - Widely spaced nipples and low hair line - Congenital cardiac defect - Club foot
Golshahi et al. (2022) [41]		Unilateral cryptophthalmos		Cutaneous syndactyly of the four limbs	-	-	Low-set umbilicus containing a single umbilical artery
Laminou et al. (2022), Nigeria [15]		Complete bilateral cryptophthalmia		Polydactyly	-	-	Nasal depression Umbilical hernia
Neri et al. (2023), Colombia [20]		Bilateral anophthalmia		Syndactyly in upper and lower limbs	-	-	- Low-set ears - Micrognathia, broad nasal bridge - Ballooning abdomen - External female genitalia with imperforate hymen - Vaginal atresia - Uterus and vagina had cystic appearance
Ramadugu et al. (2023), India [23]		Unilateral cryptophthalmos		-	- Mild hydronephrosis in the right kidney - Irregular urinary bladder walls	-	-
Bouba Traore et al. (2024), Nigeria [19]	Case 1	Bilateral cryptophthalmos		Syndactyly	-	-	- Elongated skull with part of the forehead invaded by hair - Nasal depression - Shape anomaly of the thorax
	Case 2	Bilateral cryptophthalmos		Syndactyly	-	-	- Pseudo-bald, low-set ears - Depressed nasal root - Shape anomaly of the thorax - Anal imperforation with genital anomaly
Mohamed et al. (2024), Sudan [42]		Bilateral partial cryptophthalmos		Syndactyly Overlapping fingers Rocker-bottom feet	-	-	Multiple small atrial septal defects and ventricular septal defects.
Sajoura et al. (2024), Morocco [16]		Bilateral anophthalmia		Syndactyly	-	-	- Cleft palate - Dysmorphic facies with domed forehead, hypertelorism, micrognathia, low-set ears, and short neck. - Bilateral cryptorchidism. - Triventricular hydrocephalus. - Malformative tri-ventricular hydrocephalus, hypoplasia of brainstem and cerebellum, and poly-microgyria
Mangla et al. (2024), India [10]		Cryptophthalmos		-	Renal agenesis	-	-
Present case (2025), Vietnam		Bilateral cryptophthalmos		Syndactyly on hands and feet	Absence of the right kidney	-	- Nasal depression - Low-setears and dysplasia - Cleft palate - Tracheal stenosis, glottic web - Smaller eyeball of the right eye without crystalline lens.

Autopsy was refused due to cultural constraints; Sign “-”: not mentioned. CT: computed tomography; FS: Fraser syndrome.

Since Fraser syndrome is an autosomal recessive disease, genetic counseling should be proposed for at-risk families. If both parents are carriers of a disease-causing mutation, it is recommended to inform them of the 25% risk of having an affected child for each pregnancy. A prenatal genetic diagnosis is possible if the pathogenic mutations responsible for the disease have been identified in the family. We also advise prenatal genetic screening and testing for future pregnancies for our patients. Assisted reproductive technologies could help in selecting embryos which do not carry genetic and chromosomal abnormalities [43,44]. In the case of assisted reproductive technology using in vitro fertility, a pre-implantation genetic diagnosis should be assessed. However, the long-term outcomes of offspring are a cause for concern [45,46,47]. Additionally, the risk of invasive procedures should be made known to the patient [48]. Further studies are necessary to elucidate these points.

## 4. Conclusions

In summary, Fraser syndrome is a rare autosomal recessive disorder characterized by syndactyly, cryptophthalmos, and variable abnormalities. Prenatal diagnosis by ultrasound remains challenging. Therefore, counseled ultrasound scans should be performed in cases of aroused suspicion.

## Figures and Tables

**Figure 1 diagnostics-15-01606-f001:**
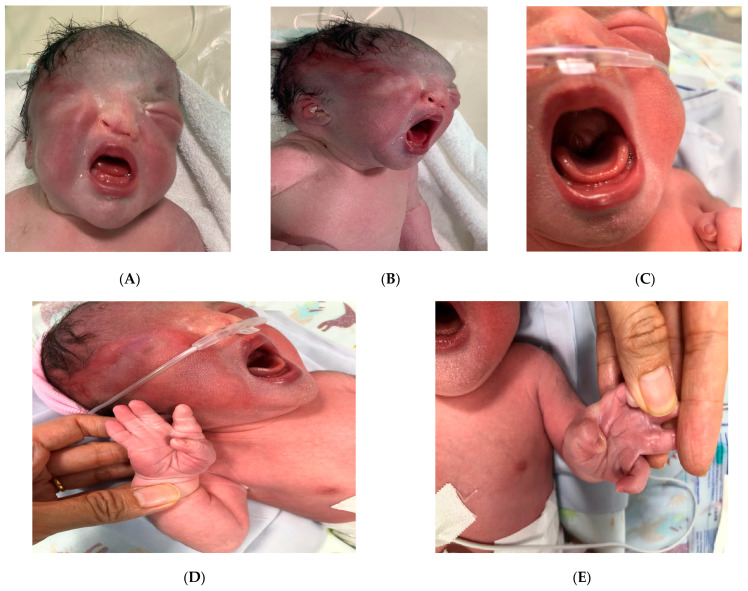
The photos of the newborn on postnatal day 1 show the face was dysmorphic with marked hypertelorism, a depressed nasal bridge, right unilateral cryptophthalmos (**A**), low-set ears and dysplasia (**B**), a cleft palate (**C**), partial III-IV-V finger membranous syndactyly on the right hand (**D**), partial II-III-IV finger membranous syndactyly on the left hand (**E**), complete II-III-IV-V foot membranous syndactyly on the right side (**F**), and complete IV-V membranous syndactyly on the left side (**G**).

**Figure 2 diagnostics-15-01606-f002:**
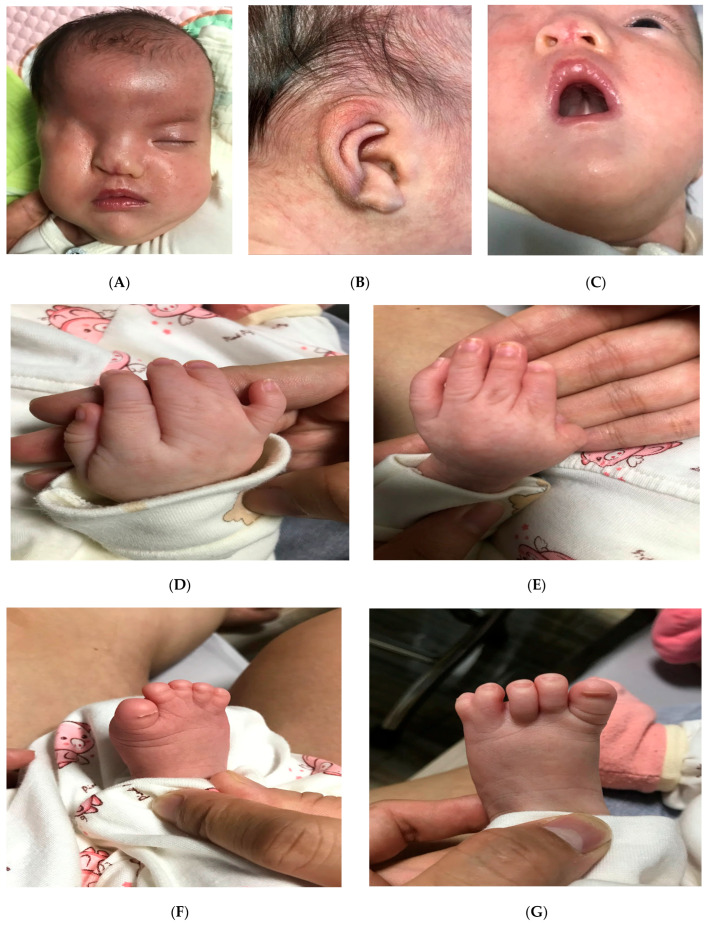
The photos of the newborn at one month and eight days showed the dysmorphic face, with marked hypertelorism, a depressed nasal bridge, right unilateral cryptophthalmos (**A**), low-set ears and dysplasia (**B**), a cleft palate (**C**), partial III-IV-V finger membranous syndactyly on the right hand (**D**), partial II-III-IV finger membranous syndactyly on the left hand (**E**), complete II-III-IV-V foot membranous syndactyly on the right side (**F**), and complete IV-V membranous syndactyly on the left side (**G**).

**Figure 3 diagnostics-15-01606-f003:**
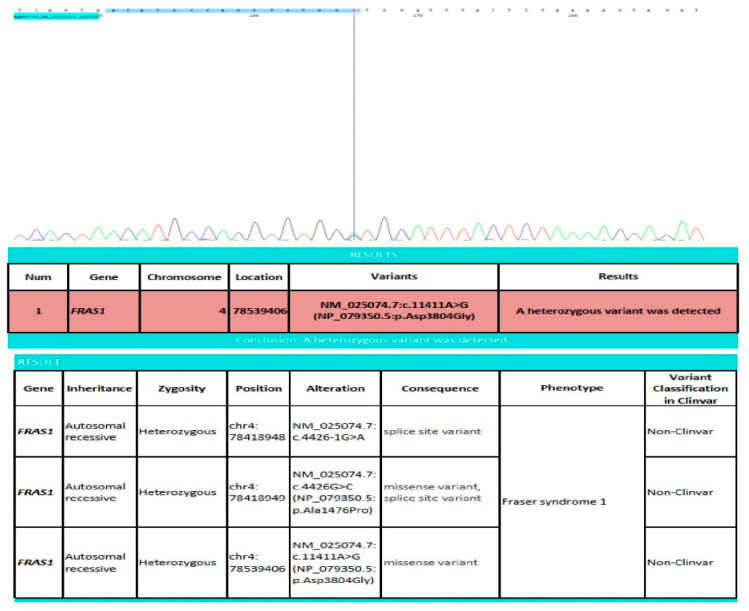
Findings showing the heterozygous variation of the FRAS1 gene of the newborn. FRAS1: Fraser extracellular matrix complex subunit 1; chr: chromosome.

**Table 1 diagnostics-15-01606-t001:** Criteria for diagnosis of Fraser syndrome.

Criteria	Thomas Criteria	Van Healst Criteria	The Present Case
**Major**	- Cryptophthalmos - Syndactyly - Abnormal genitalia - Sib with cryptophthalmos syndrome	- Syndactyly - Cryptophthalmos spectrum - Urinary tract abnormalities - Ambiguous genitalia - Laryngeal and tracheal anomalies - Positive family history	- Unilateral cryptophthalmos - Partial/complete syndactyly on fingers and toes
**Minor**	- Congenital malformation of nose - Congenital malformation of ears - Congenital malformation of larynx - Cleft lip and/or palate - Skeletal defects - Umbilical hernia - Renal agenesis - Intellectual disability	- Anorectal defects - Dysplastic ears- Skull ossification defects - Umbilical abnormalities (umbilical hernia, omphalocele, low-set umbilicus) - Nasal anomalies	- Congenital malformation of ears (low-set ears, dysplastic ears) - Depressed nasal bridge - Tracheal stenosis/vocal fold web - Cleft palate - Absence of the right kidney
**Diagnostic confirmation**	2 majors + 1 minor or 1 major + 4 minors	3 majors or 2 majors +2 minors or 1 major + 3 minors	2 majors + 5 minors

## Data Availability

No new data were created or analyzed in this study.

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
