# Peer review of "Fraser Syndrome: A Narrative Review Based on a Case from Vietnam and the Past 20 Years of Research"

_diagnostics, 2025, doi:10.3390/diagnostics15131606_

Round 1
Reviewer 1 Report
Comments and Suggestions for Authors
Esteemed Editor and author team,
I have carefully analyzed the study regarding the case presentation with Fraser syndrome as the subject and in my opinion it does not qualify for publication. The main reason is that it does not represent a progress for a better detection, knowledge or management in SF. There are many communications on this subject. There are also reviews in the specialized literature that centralize the cases up to the time of publication.
Regarding the way of presentation, this is not very accurate, the table with the cases in the literature is much too extensive and practically only the summary at the end is important.
From a medical point of view it was more important to present the diagnostic possibilities more broadly (e.g. the use of fetal MRI), it is not explained why 36 GW was chosen as the time of birth, the diagnosis of IUGR is supported but the fetus is 2700 g at 36 GW (which does not correspond to the diagnosis of IUGR) etc.
Best regards
Author Response
Reviewer 1:
I have carefully analyzed the study regarding the case presentation with Fraser syndrome as the subject and in my opinion it does not qualify for publication. The main reason is that it does not represent a progress for a better detection, knowledge or management in SF. There are many communications on this subject. There are also reviews in the specialized literature that centralize the cases up to the time of publication.
Authors responses: Thank you for your point. Fraser syndrome has been reported in literature. However, it remains an uncommon syndrome and not easy to diagnose. We wrote this paper in order to raise an awareness of the practician.
Regarding the way of presentation, this is not very accurate, the table with the cases in the literature is much too extensive and practically only the summary at the end is important.
Authors responses: We aimed to summarize the updated literature in the past-20 years to reveal some practical points and help the clinician in detecting this rare syndrome. This summary makes our paper more valuable, even it’s longer than a presentation of a case. It’s nearly a narrative review of literature based on a case. We tried to make the paper more information and helpful with tables.
From a medical point of view it was more important to present the diagnostic possibilities more broadly (e.g. the use of fetal MRI), it is not explained why 36 GW was chosen as the time of birth, the diagnosis of IUGR is supported but the fetus is 2700 g at 36 GW (which does not correspond to the diagnosis of IUGR) etc.
Authors responses: The case was induced at 38 weeks and 4 days of gestational age, not at 36 GW as we mentioned on the first paragraph of presentation case. We divided the paragraph to avoid misunderstand.
The pregnancy was diagnosed with FGR at 21 weeks and SGA fetus at 36 weeks based on Hadlock’s measurement through ultrasound following the criteria of FGR, not based on the newborn weight at birth. (Guideline No. 442: Fetal Growth Restriction: Screening, Diagnosis, and Management in Singleton Pregnancies Kingdom, JohnAshwal, EranLausman, AndreaLiauw, JessicaSoliman, NancyFigueiro-Filho, ErnestoNash, ChristopherBujold, Emmanuel Melamed, Nir et al. Journal of Obstetrics and Gynaecology Canada , Volume 45, Issue 10, 102154)
Since the patient was followed up at the private clinic room and the malformation was missed on ultrasound. Therefore, no indication of prenatal MRI in this case. However, we included your additional point in line 131: “In addition, magnetic resonance imaging (MRI) could be indicated where necessary. “
We also revised the paper for more valuable. We hope that you accepted our revised paper.
Reviewer 2 Report
Comments and Suggestions for Authors
In this article, authors described a case of Fraser syndrome and reviewed literature of the last 20 years. Good article. Clear images of the patient. Minor changes are required:
-In Table 1, in column “present case”, please MOVE “Absence of the right kidney” from “Major” to “minor” criteria, corresponding to Thomas criteria of “renal agenesis”
- In Table 1, in column “present case”, please ADD “depressed nasal bridge” (which you described under the patient’s photos, too), in row “minor criteria”, corresponding to Thomas criteria of “congenital malformation of nose”
-Then, in Table 1, if you count the total criteria this case has, please REPLACE “2 majors + 2 minors “ with “2 majors + 4 minors “
-In References, even if only 13 are recent out of 40, they are all well chosen.
Author Response
Reviewer 2:
In this article, authors described a case of Fraser syndrome and reviewed literature of the last 20 years. Good article. Clear images of the patient. Minor changes are required:
-In Table 1, in column “present case”, please MOVE “Absence of the right kidney” from “Major” to “minor” criteria, corresponding to Thomas criteria of “renal agenesis”
Authors responses: Thank you for your correction. We corrected.
- In Table 1, in column “present case”, please ADD “depressed nasal bridge” (which you described under the patient’s photos, too), in row “minor criteria”, corresponding to Thomas criteria of “congenital malformation of nose”
Authors responses: Thank you for your correction. We corrected.
-Then, in Table 1, if you count the total criteria this case has, please REPLACE “2 majors + 2 minors “ with “2 majors + 4 minors “
Authors responses: Thank you for your correction. We corrected.
-In References, even if only 13 are recent out of 40, they are all well chosen.
Authors responses: Thank you for your REVIEW.
Reviewer 3 Report
Comments and Suggestions for Authors
Fraser syndrome, a rare autosomal recessive genetic disorder, is characterized by a range of malformations, including cryptophthalmos, syndactyly, craniofacial, and genitourinary anomalies. The syndrome has an estimated prevalence of approximately 1/200 000 live births, with higher rates observed in certain populations. The syndrome arises from homozygous or compound heterozygous mutations: FRAS1, FREM2, and GRIP1, which are crucial for the development of the skin and other organs during the embryonic phase. Management of Fraser syndrome is multidisciplinary, focusing on symptomatic and supportive treatments, with the occasional need for surgical interventions in order to correct physical anomalies.
The article presents an interesting and valuable case; however, several revisions are necessary to improve clarity, consistency, and overall presentation.
The references start with (5) - line 46. Please ensure that you are following the citing guidelines throughout the manuscript.
Ensure consistent spelling of "Fraser" throughout the manuscript. It is sometimes spelled as “Frazer”.
Required revisions, based on the line #:
24-25: rephrase: “a literature review of articles from the past 20 years…”
25: replace “case” with “cases”
26: replace “relating” with “related”
28: misspelled word “ad”
30: missing abbreviation explanation for “FS”
31: “detected postnatally”
32: replace “symptoms” with “findings”
34: “at a specialised center should be recommended in suspected cases…”
56: “include”
48: missing reference.
51: replace “obligate” with “obligatory”
58-59: list all the databases you have used, without using ellipses. Avoid using ellipses throughout the manuscript
60: “case presentation”
61-69: rephrase or combine the sentences to improve readability
72: “aneuploidy”
73: “mutations”
79: missing abbreviation explanation for “FGR”
84: missing abbreviation explanation for “GA”
88: “out of our hospital” - outpatient or different hospital?
93: rephrase “The neonate was observed to have abnormal morphology.”
94: remove “a split in the roof of the mouth”
95: “abnormalities of”
100: remove “part”
109: “showed the dysmorphic face, with…”
120: “day 5 after birth”
120-121: rephrase to “The postnatal ultrasound scan demonstrated right renal agenesis.”
121: “anterior to the right iliopsoas muscle
122: “was suspected to be right renal ectopic dysplasia”
123: please rephrase - did you mean the infant gained or reached 3300g? “The infant gained/reached 3300 grams by the 28th day of life.”
127: “ocular globe”
140: “severe malformations have not been reported in association to the type of gene mutation”
142: missing comma after “reported’
144: “…show an increased risk of having an offspring with FS”
156: “did not”; “suspicions” instead of “suspects”
157: remove “therefore”
158: “was between 18-19”; “late”
159: “was diagnosed 9 months postnatally”
160: “although it can also be diagnosed during prenatal ultrasounds”
162: missing comma after “ultrasound”
167: rephrase the sentence
178: missing reference
184: avoid using the ellipsis
185: remove “hidden eye”
189: missing comma after “examination”
192: missing comma after “made”
196: “Syndactyly is defined as the fusion of two or more digits (bone or soft tissue)”
196-197: “it can be present in other syndromes, such as”...
208: “survival rates depend on”
210: “with the survival rate varying from 2 to 32 years”
212: missing comma after “abnormalities”
216: “if both parents .. mutation, it is recommended to inform them of the..”
218: “advise towards”
234: what is the purpose of this sentence?
Regarding figures and tables:
- Each figure/table should be referenced in the text, preferably right before the item, to improve readability/context.
- Consistency is important for clarity and accuracy in the tables, as well. Please ensure uniformity in how information is presented in the tables - verb tenses, formatting, phrasing
- Add a list of all abbreviations used below each table, if applicable. They are all missing some explanations (yo, wks, d)
- Replace “Italia” with “Italy”; “Roumania” with “Romania”
- Format the tables to landscape form to improve readability
- Throughout Table 2: survival “of” 2 years
- Saleem et al (2015): 2kg2 (remove the second “2”)
- Abdal case 3: “antenatal” is misspelled
- Throughout the table 2: replace “terme” with “term”; “primipare” with “primipara”; dead” with “death”
- Table 3: replace “liquor” with “amniotic fluid”; Replace “numerous ascites” with “high volume of ascites”
- Table 5 could be a supplemental item. Summarize the findings within the text.
You could add more epidemiological information, risk factors, and clinical presentation/diagnosis information.
Author Response
Reviewer 3:
Fraser syndrome, a rare autosomal recessive genetic disorder, is characterized by a range of malformations, including cryptophthalmos, syndactyly, craniofacial, and genitourinary anomalies. The syndrome has an estimated prevalence of approximately 1/200 000 live births, with higher rates observed in certain populations. The syndrome arises from homozygous or compound heterozygous mutations: FRAS1, FREM2, and GRIP1, which are crucial for the development of the skin and other organs during the embryonic phase. Management of Fraser syndrome is multidisciplinary, focusing on symptomatic and supportive treatments, with the occasional need for surgical interventions in order to correct physical anomalies.
The article presents an interesting and valuable case; however, several revisions are necessary to improve clarity, consistency, and overall presentation.
The references start with (5) - line 46. Please ensure that you are following the citing guidelines throughout the manuscript.
Authors responses: Thank you for your correction. We corrected.
In the text, reference numbers were placed in square brackets [ ] and placed before the punctuation; for example [1], [1–3] or [1,3].
Ensure consistent spelling of "Fraser" throughout the manuscript. It is sometimes spelled as “Frazer”.
Authors responses: Thank you for your correction. We corrected.
Required revisions, based on the line #:
24-25: rephrase: “a literature review of articles from the past 20 years…”
Authors responses: Thank you for the point. We corrected it.
25: replace “case” with “cases”
Authors responses: Thank you for the point. We corrected it.
26: replace “relating” with “related”
Authors responses: Thank you for the point. We corrected it.
28: misspelled word “ad”
Authors responses: We deleted it.
30: missing abbreviation explanation for “FS”
Authors responses: Thank you for the point. We corrected it.
31: “detected postnatally”
Authors responses: Thank you for the point. We corrected it.
32: replace “symptoms” with “findings”
Authors responses: Thank you for the point. We corrected it.
34: “at a specialised center should be recommended in suspected cases…”
Authors responses: Thank you for the point. We corrected it.
56: “include”
Authors responses: Thank you for the point. We corrected it.
48: missing reference.
Authors responses: Thank you for the point. We corrected it.
51: replace “obligate” with “obligatory”
Authors responses: Thank you for the point. We corrected it.
58-59: list all the databases you have used, without using ellipses. Avoid using ellipses throughout the manuscript
Authors responses: Thank you for the point. We corrected it.
60: “case presentation”
Authors responses: Thank you for the point. We corrected it.
61-69: rephrase or combine the sentences to improve readability
Authors responses: Thank you for the point. We modified it.
72: “aneuploidy”
Authors responses: Thank you for the point. We corrected it.
73: “mutations”
Authors responses: Thank you for the point. We corrected it.
79: missing abbreviation explanation for “FGR”
Authors responses: Thank you for the point. We corrected it.
84: missing abbreviation explanation for “GA”
Authors responses: Thank you for the point. We corrected it.
88: “out of our hospital” - outpatient or different hospital?
Authors responses: Thank you for the point. We corrected it: “at the different hospital”
93: rephrase “The neonate was observed to have abnormal morphology.”
Authors responses: Thank you for the point. We corrected it.
94: remove “a split in the roof of the mouth”
Authors responses: Thank you for the point. We removed it.
95: “abnormalities of”
Authors responses: Thank you for the point. We corrected it.
100: remove “part”
Authors responses: Thank you for the point. We corrected it.
109: “showed the dysmorphic face, with…”
Authors responses: Thank you for the point. We corrected it.
120: “day 5 after birth”
Authors responses: Thank you for the point. We corrected it.
120-121: rephrase to “The postnatal ultrasound scan demonstrated right renal agenesis.”
Authors responses: Thank you for the point. We corrected it.
121: “anterior to the right iliopsoas muscle
Authors responses: Thank you for the point. We corrected it.
122: “was suspected to be right renal ectopic dysplasia”
Authors responses: Thank you for the point. We corrected it.
123: please rephrase - did you mean the infant gained or reached 3300g? “The infant gained/reached 3300 grams by the 28th day of life.”
Authors responses: Thank you for the point. We corrected it.” The infant gained 3300 grams by the 28th day of life.”
127: “ocular globe”
Authors responses: Thank you for the point. We corrected it.
140: “severe malformations have not been reported in association to the type of gene mutation”
Authors responses: Thank you for the point. We corrected it.
142: missing comma after “reported’
Authors responses: Thank you for the point. We corrected it.
144: “…show an increased risk of having an offspring with FS”
Authors responses: Thank you for the point. We corrected it.
156: “did not”; “suspicions” instead of “suspects”
Authors responses: Thank you for the point. We corrected it.
157: remove “therefore”
Authors responses: Thank you for the point. We removed it.
158: “was between 18-19”; “late”
Authors responses: Thank you for the point. We corrected it.
159: “was diagnosed 9 months postnatally”
Authors responses: Thank you for the point. We corrected it.
160: “although it can also be diagnosed during prenatal ultrasounds”
Authors responses: Thank you for the point. We corrected it.
162: missing comma after “ultrasound”
Authors responses: Thank you for the point. We corrected it.
167: rephrase the sentence
Authors responses: Thank you for the point. We modified it.
178: missing reference
Authors responses: Thank you for the point. We added it.
184: avoid using the ellipsis
Authors responses: Thank you for the point. We corrected it.
185: remove “hidden eye”
Authors responses: Thank you for the point. We corrected it.
189: missing comma after “examination”
Authors responses: Thank you for the point. We corrected it.
192: missing comma after “made”
Authors responses: Thank you for the point. We corrected it.
196: “Syndactyly is defined as the fusion of two or more digits (bone or soft tissue)”
Authors responses: Thank you for the point. We corrected it.
196-197: “it can be present in other syndromes, such as”...
Authors responses: Thank you for the point. We corrected it.
208: “survival rates depend on”
Authors responses: Thank you for the point. We corrected it.
210: “with the survival rate varying from 2 to 32 years”
Authors responses: Thank you for the point. We corrected it.
212: missing comma after “abnormalities”
Authors responses: Thank you for the point. We corrected it.
216: “if both parents .. mutation, it is recommended to inform them of the..”
Authors responses: Thank you for the point. We corrected it.
218: “advise towards”
Authors responses: Thank you for the point. We corrected it.
234: what is the purpose of this sentence?
It was explained for the definition of severe neonatal outcomes in table 5. But the table is moved, thus it make the sentence confusing. We change it to the supplemental Table 1 now following your suggestion.
Regarding figures and tables:
- Each figure/table should be referenced in the text, preferably right before the item, to improve readability/context.
Authors responses: Thank you for the point. We corrected it.
- Consistency is important for clarity and accuracy in the tables, as well. Please ensure uniformity in how information is presented in the tables - verb tenses, formatting, phrasing
Authors responses: Thank you for the point. We corrected it.
- Add a list of all abbreviations used below each table, if applicable. They are all missing some explanations (yo, wks, d)
- Authors responses: Thank you for the point. We corrected it.
- Replace “Italia” with “Italy”; “Roumania” with “Romania”
Authors responses: Thank you for the point. We corrected it.
- Format the tables to landscape form to improve readability
Authors responses: Thank you for the point. We corrected it. We hope the production team could help us in editing the table better.
- Throughout Table 2: survival “of” 2 years
Authors responses: Thank you for the point. We corrected it.
- Saleem et al (2015): 2kg2 (remove the second “2”)
Authors responses: Thank you for the point. We corrected it.
Abdal case 3: “antenatal” is misspelled
- Authors responses: Thank you for the point. We corrected it.
- Throughout the table 2: replace “terme” with “term”; “primipare” with “primipara”; dead” with “death”
Authors responses: Thank you for the point. We corrected it.
- Table 3: replace “liquor” with “amniotic fluid”; Replace “numerous ascites” with “high volume of ascites”
Authors responses: Thank you for the point. We corrected it.
- Table 5 could be a supplemental item. Summarize the findings within the text.
Authors responses: Thank you for the point. We move Table 5 to supplemental file.
You could add more epidemiological information, risk factors, and clinical presentation/diagnosis information.
Authors responses: Thank you for the point. We added it.
Reviewer 4 Report
Comments and Suggestions for Authors
The present manuscript reports of a newborn with Fraser syndrome. Fraser syndrome is a rare genetic condition (prevalence 1 in 200 000) characterized by cryptophthalmos, cutaneous syndactyly, and genitourinary anomalies. Other tissues and organs can also be affected.
I have the following recommendations:
Introduction - this section could be expanded. Consanguineous marriage is not a risk factor for Fraser syndrome, it could increase the risk for autosomal recessive conditions in general.
Case presentation, lines 72-77 - did you perform the tests for these genetic disorders on the parents, it is not clear?
Line 82 - the amount of the amniotic fluid was below the normal one, this should be indicated in the text.
Lines 127 - 134 - If the patient has a compound heterozygous genotype, both alleles must have different pathogenic variants, typically one inherited from each parent. However, the text states: One heterozygous variant is found in the patient. The father has one heterozygous variant. The mother has a normal genotype. This contradicts the claim of compound heterozygosity — unless there’s an error in reporting the mother’s genotype.
Discussion, lines 142 - 147 - poor antenatal care is not a risk factor for Fraser syndrome, this is an autosomal recessive condition. The recurrence risk is 25% for each pregnancy, it is a matter of chance whether it is gonna be the first pregnancy, the ninth or no affected baby at all.
Lines 148-166 - there is not a single reference for the whole paragraph. In the next paragraph there is only one reference - number 20.
It would be interesting to share any details on the development of the child, did you perform any surgical operation on the child, did you perform scintigraphy of the kidney?
Table 2 - most of the described cases do not have a molecular-genetic confirmation, this is a limitation of the table.
Minor recommendations - check the abbreviations, not all are listed in the text, some grammatical mistakes.
Comments on the Quality of English Language
Revise the English language.
Author Response
Reviewer 4:
The present manuscript reports of a newborn with Fraser syndrome. Fraser syndrome is a rare genetic condition (prevalence 1 in 200 000) characterized by cryptophthalmos, cutaneous syndactyly, and genitourinary anomalies. Other tissues and organs can also be affected.
I have the following recommendations:
Introduction - this section could be expanded. Consanguineous marriage is not a risk factor for Fraser syndrome; it could increase the risk for autosomal recessive conditions in general.
Authors responses: We added the point. Thank you for your suggestion.
Case presentation, lines 72-77 - did you perform the tests for these genetic disorders on the parents, it is not clear?
Authors responses: We perform the genetic tests by maternal plasma. We added it.
Line 82 - the amount of the amniotic fluid was below the normal one, this should be indicated in the text.
Authors responses: The amniotic fluid is defined as oligoamnios if the AFI (amniotic fluid index below 5cm or the maximum vertical pocket under 2 cm). In our case, the AFI is 6 cm. Thus, we did not analyze deeply this point. But we should mention its value in the presentation case.
Lines 127 - 134 - If the patient has a compound heterozygous genotype, both alleles must have different pathogenic variants, typically one inherited from each parent. However, the text states: One heterozygous variant is found in the patient. The father has one heterozygous variant. The mother has a normal genotype. This contradicts the claim of compound heterozygosity — unless there’s an error in reporting the mother’s genotype.
Authors responses: FRS is genetically heterogeneous. Our case is not the first case related to this genetic abnormality of the parents (Figure 3 and Supplemental Figure 1A-B).
Slavotinek A, Li C, Sherr EH, Chudley AE. Mutation analysis of the FRAS1 gene demonstrates new mutations in a propositus with Fraser syndrome. Am J Med Genet A. 2006 Sep 15;140(18):1909-14. doi: 10.1002/ajmg.a.31399. PMID: 16894541.
Hoefele J, Wilhelm C, Schiesser M, Mack R, Heinrich U, Weber LT, Biskup S, Daumer-Haas C, Klein HG, Rost I. Expanding the mutation spectrum for Fraser syndrome: identification of a novel heterozygous deletion in FRAS1. Gene. 2013 May 15;520(2):194-7. doi: 10.1016/j.gene.2013.02.031. Epub 2013 Mar 6. PMID: 23473829.
Discussion, lines 142 - 147 - poor antenatal care is not a risk factor for Fraser syndrome, this is an autosomal recessive condition. The recurrence risk is 25% for each pregnancy, it is a matter of chance whether it is gonna be the first pregnancy, the ninth or no affected baby at all.
Authors responses: Thank you for your point. We modified it.
Lines 148-166 - there is not a single reference for the whole paragraph. In the next paragraph there is only one reference - number 20.
Authors responses: Thank you for your point. We added the ref [1] and [8]. The next points were mentioned in table 3-4.
It would be interesting to share any details on the development of the child, did you perform any surgical operation on the child, did you perform scintigraphy of the kidney?
Authors responses: Thank you for your point.
As we mentioned in the discussion: “Following functional stability, the esthetics are concerned. The management requires many step-by-step surgeries [2]. The surgical repair of FS requires a multidisciplinary team (including maxillofacial surgeon, ear-nose throat (ENT) specialist, nephrologist, ophthalmologist, and other specialists) [30,31].”
The baby is 5-month-old now. Thus, the surgery remains early. We will do it in the future.
Line 99-100: “The baby was monitored for a systematic approach of surgical management in the future.”
Table 2 - most of the described cases do not have a molecular-genetic confirmation, this is a limitation of the table.
Authors responses: As we mentioned in the paper, almost all cases were related to the poor condition of family. Thus, the assessment of genetic analysis is limited. That’s also a limitation of literature. In our case, since the parents are medical staff, they understand and accept for genetic analysis after birth.
Minor recommendations - check the abbreviations, not all are listed in the text, some grammatical mistakes.
Authors responses: Thank you for your review. We corrected it.
Round 2
Reviewer 1 Report
Comments and Suggestions for Authors
Esteemed Editor and author team,
I have analyzed the changes made by the authors. Following the evaluation of the study, I maintain my opinion that the article does not qualify for publication.
Best regards
Author Response
I have analyzed the changes made by the authors. Following the evaluation of the study, I maintain my opinion that the article does not qualify for publication.
Authors responses: We are sorry to disappoint you. Thank you for your review time.
We tried to revise the paper again.
We hope you reconsider our paper.
Reviewer 3 Report
Comments and Suggestions for Authors
The first cited references are [5,6]. They should be mentioned in the order of appearance, starting with [1] - which means you will probably be required to redo the citations throughout the manuscript.
Page 2: "...rubella infections"
Figure 2 should appear after being cited in the manuscript.
Page 9: "selecting the embryos which do not carry the..."; "the long-term outcomes of the offsprings should be a cause for concern"
Table 4 is missing the list of abbreviations below it.
Please add a list of all the abbreviations used throughout the manuscript, preferably in alphabetical order, at the beginning/end of the manuscript.
Author Response
The first cited references are [5,6]. They should be mentioned in the order of appearance, starting with [1] - which means you will probably be required to redo the citations throughout the manuscript.
Authors responses: Thank you again for this point. We missed this paragraph cited by ref [1-4] in the submitted manuscript. We added it.
Page 2: "...rubella infections"
Authors responses: Thank you for this correction. We modified it.
Figure 2 should appear after being cited in the manuscript.
Authors responses: Thank you for this point. We changed it.
Page 9: "selecting the embryos which do not carry the..."; "the long-term outcomes of the offsprings should be a cause for concern"
Authors responses: Thank you for this correction. We modified it.
Table 4 is missing the list of abbreviations below it.
Authors responses: Thank you for this correction. We added it.
Please add a list of all the abbreviations used throughout the manuscript, preferably in alphabetical order, at the beginning/end of the manuscript.
Authors responses: Thank you for this point. We added it.
Reviewer 4 Report
Comments and Suggestions for Authors
Dear Authors,
Thank you for your submission. I appreciate your effort in presenting this case; however, I recommend consulting a clinical geneticist when preparing case reports involving genetic diagnoses, as there appear to be misunderstandings of some fundamental genetic concepts.
Specifically, the term compound heterozygous refers to an individual carrying two different pathogenic variants at the same locus—typically, one on each allele. In your case, only a single heterozygous variant is reported. According to the OMIM database, Fraser syndrome is inherited in an autosomal recessive manner, which typically requires biallelic pathogenic variants for a molecular diagnosis. The absence of a second variant raises important questions.
Furthermore, the report states that the mother is not a carrier, which also calls for clarification. This discrepancy could potentially be explained by limitations of the genetic testing method used—for instance, a second variant could be located in a non-coding region or be a structural variant not detectable by the analysis applied.
Until a second pathogenic variant is identified, the diagnosis remains clinical, not molecularly confirmed, which impacts the overall interpretation and significance of the case.
Author Response
Thank you for your submission. I appreciate your effort in presenting this case; however, I recommend consulting a clinical geneticist when preparing case reports involving genetic diagnoses, as there appear to be misunderstandings of some fundamental genetic concepts.
Specifically, the term compound heterozygous refers to an individual carrying two different pathogenic variants at the same locus—typically, one on each allele. In your case, only a single heterozygous variant is reported. According to the OMIM database, Fraser syndrome is inherited in an autosomal recessive manner, which typically requires biallelic pathogenic variants for a molecular diagnosis. The absence of a second variant raises important questions.
Furthermore, the report states that the mother is not a carrier, which also calls for clarification. This discrepancy could potentially be explained by limitations of the genetic testing method used—for instance, a second variant could be located in a non-coding region or be a structural variant not detectable by the analysis applied.
Until a second pathogenic variant is identified, the diagnosis remains clinical, not molecularly confirmed, which impacts the overall interpretation and significance of the case.
Authors response: Thank you for your point. We had a mistake error when preparing the revised manuscript. The mother carried also a heterozygous variant relating to FRAS1 gene. We corrected carefully through the revised paper.
Line 119-121: The parental genetic analysis of revealed a heterozygous variant of FRAS1 gene on chromosome 4, location 78418948 relating to variation of NM_025074.7:c.4426-1G>A (Father) and a heterozygous variant of FRAS1 gene on chromosome 4, location 78539406 relating to variation of NM_025074.7:c.11411A>G (Mother) (Supplemental Figure 1A-B).
Round 3
Reviewer 3 Report
Comments and Suggestions for Authors
Please revise the list of abbreviations. There are still some that are missing (such as CT, ENT, CMV, NIPT...) and some that are not correctly added (FR instead of FS). Add all the abbreviations used in the manuscript, including the ones from the tables/figures.
Cite the Thomas et al article you have mentioned in the first paragraph of the introduction.
Author Response
Thank you for your careful revision again. We have updated all your requirement regarding abbreviations. Please check the red track changes.
Regarding the cited reference of Thomas et., we would like to explain the point concerning the discovery of disease was extracted from the review in reference 3. Thus, cited reference 3 is more suitable.
Bouaoud J, Olivetto M, Testelin S, Dakpe S, Bettoni J, Devauchelle B. Fraser syndrome: review of the literature illustrated by a historical adult case. Int J Oral Maxillofac Surg. 2020 Oct;49(10):1245-1253. doi: 10.1016/j.ijom.2020.01.007. Epub 2020 Jan 22. PMID: 31982235.
Reviewer 4 Report
Comments and Suggestions for Authors
Dear Authors,
I was quite surprised by the recent changes made to the manuscript, particularly the sudden identification of the index patient’s mother as a heterozygous carrier. This is especially unexpected given that this is the second time I am raising this point. In your previous response to my comments on the same issue, you stated, and I quote: “Authors’ response: FRS is genetically heterogeneous. Our case is not the first case related to this genetic abnormality of the parents (Figure 3 and Supplemental Figure 1A-B).” Given this, the current claim that the mother is now a confirmed carrier raises concerns. I must express my reservations about the consistency and reliability of your case report.
Author Response
We are sorry for this confusion. After submitting the first revision, we have checked and found that we missed the second page of maternal genetic analysis. We have contacted the editors to updated the revision version by email. However, it was late. Thus, it made you feel confused. We send you again the evidence of email and we also presented the genetic result from the mother.
